# Protein Dielectrophoresis: I. Status of Experiments and an Empirical Theory

**DOI:** 10.3390/mi11050533

**Published:** 2020-05-22

**Authors:** Ralph Hölzel, Ronald Pethig

**Affiliations:** 1Fraunhofer Institute for Cell Therapy and Immunology, Branch Bioanalytics and Bioprocesses (IZI-BB), Am Mühlenberg 13, 14476 Potsdam-Golm, Germany; ralph.hoelzel@izi-bb.fraunhofer.de; 2School of Engineering, Institute for Integrated Micro and Nanosystems, University of Edinburgh, The King’s Buildings, Edinburgh EH9 3JF, UK

**Keywords:** Clausius–Mossotti function, dielectrophoresis, dielectric spectroscopy, interfacial polarization, proteins

## Abstract

The dielectrophoresis (DEP) data reported in the literature since 1994 for 22 different globular proteins is examined in detail. Apart from three cases, all of the reported protein DEP experiments employed a gradient field factor $\nabla E_{m}^{2}$ that is much smaller (in some instances by many orders of magnitude) than the ~4 × 10^21^ V^2^/m^3^ required, according to current DEP theory, to overcome the dispersive forces associated with Brownian motion. This failing results from the macroscopic Clausius–Mossotti (*CM*) factor being restricted to the range 1.0 > *CM* > −0.5. Current DEP theory precludes the protein’s permanent dipole moment (rather than the induced moment) from contributing to the DEP force. Based on the magnitude of the β-dispersion exhibited by globular proteins in the frequency range 1 kHz–50 MHz, an empirically derived molecular version of *CM* is obtained. This factor varies greatly in magnitude from protein to protein (e.g., ~37,000 for carboxypeptidase; ~190 for phospholipase) and when incorporated into the basic expression for the DEP force brings most of the reported protein DEP above the minimum required to overcome dispersive Brownian thermal effects. We believe this empirically-derived finding validates the theories currently being advanced by Matyushov and co-workers.

## 1. Introduction

Dielectrophoresis (DEP) studies of biological particles have progressed from the microscopic scale of cells and bacteria, through the much smaller scale of virions to the molecular scale of DNA and proteins [1]. In a pioneering study of 1994, Washizu et al. [2] demonstrated that DEP forces capable of overcoming randomizing Brownian influences could be exerted on protein molecules (avidin, chymotripsinogen, concanavalin and ribonuclease) using micrometer-sized electrodes. The applied fields (0.4–1.0 × 10^6^ V/m) were considered to be substantially lower than standard DEP theory predicts [2]. In fact, the word ‘substantially’ can be considered as an understatement of the situation. As reviewed elsewhere, at least 22 different globular proteins have now been investigated for their DEP responses [3,4,5,6,7]. In all the analyses by the authors of the cited studies, the so-called Clausius–Mossotti (*CM*) function has been invoked. However, the macroscopic electrostatic concepts and assumptions used in the theoretical derivation of *CM* arguably fail to describe the situation for nanoparticles, such as proteins, that possess a permanent dipole moment, interact with water dipoles of hydration, and possess other physico-chemical attributes at the molecular scale [6,7,8]. The fact that standard DEP theory does not provide a basis for understanding protein DEP is recognized as “a well-accepted paradigm, repeated in numerous studies” [6]. In another recent review it is correctly stated that protein DEP remains under development because due to their small size proteins “require greater magnitudes of electric field gradients to achieve manipulation” [7]. Put more bluntly, protein DEP is considered to not have a theoretical leg to stand on! 

A new theory is in fact evolving in terms of a description at the molecular level of how a macroscopic dielectric sample responds to an applied electric field [8,9,10]. This involves a consideration of the actual ‘cavity field’ experienced by the protein molecule, as well as the time-dependent correlation of the total electric moment of the protein. This moment is a resultant of all the permanent and induced moments of the system comprising the protein molecule’s polypeptide chain(s), the protein’s hydration sheath, as well as neighboring water molecules under the electrostatic influence of the protein’s induced and permanent dipole field. 

The purpose of this and an accompanying paper [11] is to critically evaluate the protein DEP literature, to derive an empirical-based theory, and to then describe and summarize the molecular-based theory developed by Matyushov and colleagues [8,10]. In this paper we examine aspects of the reported protein DEP work not covered in previous reviews, and conclude that the reported DEP responses for a range of proteins are largely consistent. Practically all of the DEP data cannot be explained in terms of the induced-dipole moment theory currently employed by the DEP community. The previous proposal [9] that the permanent, intrinsic, dipole moment of a protein, manifested when polarized as a dielectric β-dispersion, should form the underlying basis for a proper theory of protein DEP is repeated here. It is also shown that the reported DEP responses of protein molecules are understandable if the ‘cavity’ field experienced by the protein is at least 1000-times larger than the local macroscopic field in the surrounding aqueous medium. By linking the β-dispersion (a molecular-scale phenomenon) to the macroscopic phenomenon known as the Maxwell-Wagner interfacial polarization exhibited by colloids, we derive an empirical relationship to describe this amplification of the protein’s cavity field. This empirical relationship underscores the fact that the macroscopic *CM* function employed in the present standard DEP theory is an analogue of (but not the same as) the molecular *CM*-relation that formed the bedrock of classical dielectric theory [12] used to describe the electrical polarization of proteins [13]. However, to exploit the potential benefits that protein DEP can offer to basic research needs and clinical applications [6], we require a solid molecular-based theory. In our opinion, the most promising theory currently being developed for protein DEP is that emerging from Matyushov’s group [8,10]. An attempt to summarize this is given in the accompanying paper [11], within the frameworks of the development and application of the molecular *CM*-relation in classical dielectric theory; the key dielectric properties of solvated proteins; the published work on protein DEP. 

In all of this it is instructive to appreciate how the *CM*-factor is incorporated into present DEP theory. A detailed description is presented elsewhere [9], but in brief it is based on the following sequence of assumptions and derivations: (i)The internal electrical field *E**_i_* induced in an uncharged (or uniformly charged) spherical particle, of radius *R*, located in an electric field *E**_m_* within a dielectric medium is given by:
(1)$$E_{i}=\left(\frac{3\varepsilon_{m}}{\varepsilon_{p}+2\varepsilon_{m}}\right)E_{m}$$
with *ɛ**_p_* and *ɛ**_m_* the relative permittivity of the particle and surrounding medium, respectively. It is assumed that *ɛ**_p_* and *ɛ**_m_* are well defined. At the molecular scale this requires certain conditions to be met regarding dipole–dipole correlations. Boundary conditions also assume that the electric potential, current density and displacement flux are continuous across an infinitesimally thin surface at the sphere’s interface with the surrounding medium. Fine details such as those that occur, for example, at the molecular interface between a protein and its hydration sheath are not considered.
(ii)The induced polarization *P_p_* per unit volume of the sphere is given by:
(2)$$P_{p}=\left(\varepsilon_{p}-\varepsilon_{m}\right)\varepsilon_{o}E_{i}=3\varepsilon_{o}\varepsilon_{m}\left(\frac{\varepsilon_{p}-\varepsilon_{m}}{\varepsilon_{p}+2\varepsilon_{m}}\right)E_{m}$$
where *ɛ**_o_* is the permittivity of vacuum. The macroscopic dielectric concepts involved in this equation and throughout this paper are summarized in Figure 1. It is assumed that the polarization *P**_m_* of the surrounding medium remains uniform right up to the particle–medium interface. This assumption requires examination at the molecular scale.(iii)The dipole moment *m* of the sphere is the value of *P_p_* multiplied by the sphere’s volume:
(3)$$m=4\pi R^{3}\varepsilon_{o}\varepsilon_{m}\left(\frac{\varepsilon_{p}-\varepsilon_{m}}{\varepsilon_{p}+2\varepsilon_{m}}\right)E_{m}$$
The term in brackets in Equations (2) and (3) is the Clausius–Mossotti (*CM*) function. Depending on the relative values of *ɛ**_p_* and *ɛ**_m_*, *CM* is limited to values between +1.0 (*ɛ**_p_*
_>>_
*ɛ**_m_*) and −0.5 (*ɛ**_p_*
_<<_
*ɛ**_m_*). This represents a severe limitation, at the macroscopic scale, to the range of effective dipole moment densities that a particle can assume. (iv)For the case where *E**_m_* has a gradient, the particle experiences a DEP force given by:
(4)$$F_{DEP}=\left(m\cdot \nabla \right)E_{m}$$
where ∇ is the gradient (del) operator and *E**_m_* is assumed irrotational (i.e., ∇×*E**_m_* = 0). This assumption holds if *E**_m_* is said to be a conservative field. In our particular case of DEP, this means that moving a polarized particle from location *a* to *b*, and then back again to location *a*, will involve no net expenditure of work by the field. The actual path taken in moving from say *a* to *z* is of no relevance. In the language of thermodynamics each infinitesimal change in location is reversible. At the molecular level, the DEP motion of a protein involves the breaking (enthalpy absorbed and entropy increased) and remaking of hydrogen-bonded water networks at the hydrodynamic plane of shear. Some interesting variations of changes in Gibbs free energy (ΔG = ΔH − T ΔS) might occur. The response of an assembly of dipoles to an external electric field is basically a thermodynamically non-equilibrium process—the thermal energy is never equally distributed among the various degrees of motional freedom of the dipoles. Perhaps, at the molecular level, each infinitesimal change in location is not reversible?


From Equations (3) and (4):(5)$$F_{DEP}=4\pi R^{3}\varepsilon_{o}\varepsilon_{m}\left(\frac{\varepsilon_{p}-\varepsilon_{m}}{\varepsilon_{p}+2\varepsilon_{m}}\right)\left(E_{m}\cdot \nabla \right)E_{m}=2\pi R^{3}\varepsilon_{o}\varepsilon_{m}\left(\frac{\varepsilon_{p}-\varepsilon_{m}}{\varepsilon_{p}+2\varepsilon_{m}}\right)\nabla E_{m}^{2}$$

Equation (5) can be extended to describe oblate and prolate spheroids by introducing a polarization parameter that moderates the internal field, and AC fields are accommodated using a complex *CM* (i.e., contains real and imaginary components) that takes into account the phase difference between charge displacement and ohmic currents in particles exhibiting dielectric losses. The complex conductivity and permittivity are related by $\sigma^{*}=i\omega \varepsilon_{o}\varepsilon^{*}$ where *i* = √−1 and *ω* is the radian frequency of the applied r.m.s. field *E_o_*. The form of *CM* (the term in brackets) shown in Equation (5) is valid at high frequencies (typically >50 MHz). At DC and below ~100 Hz
(6)$$CM=\left(\frac{\sigma_{p}-\sigma_{m}}{\sigma_{p}+2\sigma_{m}}\right)$$

At intermediate frequencies *CM* contains real and imaginary components, with only the real value (Re[*CM*]) employed in Equation (5). 

## 2. The Basic Problem to Be Empirically Resolved 

According to the standard induced dipole moment model of DEP, *CM* is limited to the range of values 1.0 > *CM* > −0.5, and so the parameters that predominantly determine the magnitude of *F_DEP_* are particle size and the magnitude of the field-parameter $\nabla E_{m}^{2}$. In a first approximation a cubic root relation is expected regarding the physical dimensions of a globular protein and its molecular weight. However, an empirical relationship, provided by Malvern Panalytical^®^ in their calculator software, gives a good estimate of a protein’s hydrodynamic (Stokes) radius. This relationship is plotted in Figure 2, to show that those proteins reported to exhibit DEP responses have radii in the range 2–7 nm. Values of *E_m_* and of $\nabla E_{m}^{2}$ in the ranges 10^5^–10^8^ V/m and 10^12^–10^24^ V^2^/m^3^, respectively, are reported for the DEP translocation and trapping of protein molecules. We can ask to what extent these fields and their gradients are consistent with the expectations of the current theoretical model of DEP when applied to a globular protein molecule. This question can be addressed by considering both the time-averaged free energy (*U_DEP_* = −(*mE_m_*)/2) of an electrically polarized particle and the work required to overcome the maximum dispersive (diffusional) force acting on it [4] (pp. 352–353). The first approach addresses the extent to which *U_DEP_* represents a sufficiently deep ‘trap’ to compete against thermal energy (3*kT*)/2 associated with Brownian motion. Using the relationship given for the dipole moment *m* in Equation (3): (7)$$U_{DEP}=-\frac{1}{2}m\cdot E_{m}=-2\pi R^{3}\varepsilon_{o}\varepsilon_{m}\left[CM\right]E_{m}^{2}$$

The total free energy *U_T_* of a polarized protein molecule is the sum of the Brownian thermal energy and *U_DEP_*: (8)$$U_{T}=\frac{3}{2}kT+U_{DEP}=\frac{3}{2}kT-2\pi R^{3}\varepsilon_{o}\varepsilon_{m}\left[CM\right]E_{m}^{2}$$

For the protein to be trapped by DEP, *U_T_* must have a negative value—it should represent a sufficiently deep potential energy well for the molecule to be trapped for a time equivalent to the inverse of its probability to escape. For proteins such as bovine serum albumin (BSA) and avidin (*R* ≈ 3.5 nm, *T* = 300 K, and assigning *CM* = 0.5) suspended in an aqueous medium (i.e., *ɛ_m_* ≈ 78) the required field is *E**_m_* ≥ 2.3 × 10^7^ V/m. The maximum dispersive force per protein molecule is equal to -*kT*/2*R*. For *R* ≈ 3.5 nm and *T* = 300 K, this force is 1.2 × 10^−12^ N. For the protein molecule to exhibit DEP it must oppose this dispersive force. With the expression for *F_DEP_* from Equation (5) or from Equation (7), using the definition $F_{DEP}=-\nabla U_{DEP}$, this implies the following condition must hold: (9)$$2\pi R^{3}\varepsilon_{o}\varepsilon_{m}\left[CM\right]\nabla E_{m}^{2}>1.2\times 10^{-12}N$$

For *R* = 3.5 nm and *CM* = 0.5 this requires $\nabla E_{m}^{2}$ > 3.5 × 10^21^ V^2^/m^3^. As discussed in Section 3.7, only two of the reported values of $\nabla E_{m}^{2}$ have exceeded this minimum value. In one reported DEP manipulation of BSA a value of 10^12^ V^2^/m^3^ is cited! There are also the interesting cases where both positive and negative DEP of BSA have been reported at DC and 1 kHz, and where DEP of opposite polarities have also been reported for the same protein types at DC. 

Considering the potential importance that protein DEP can offer, these experimental quirks should be addressed by a critical evaluation of both the validity of Equation (5) for protein DEP and the reported studies themselves. An effort is made here to examine, in broader detail than we consider has been attempted previously by others, the reported DEP literature on proteins and the relevant theory. An assessment is made of possible confounding influences, such as protein aggregation. All of the reported studies of protein DEP appear to be the results of careful work, and so even the more puzzling cases should assist in a better understanding of protein DEP and for the development of a more appropriate theoretical model to describe the DEP of proteins. We show that the protein DEP results reported to date are consistent with a model in which an evaluation of an induced dipole moment through Equations (1)–(3) should not be treated as the sole pertinent consideration. An important step forward regarding Equation (5) should be inclusion of the intrinsic (i.e., permanent) dipole moment possessed and well-studied for globular proteins [13]. Of particular importance is the orientation polarization of this dipole moment [8,10]—a feature overlooked in a previous discourse where the protein was considered to be a rigid dipole [9]. 

## 3. The Status of Protein Dielectrophoresis (DEP) Experimentation

### 3.1. Summary of Protein DEP 

The most studied protein for its DEP characteristics is BSA. Figure 3 and Figure 4 provide summaries of the observed DEP polarity, the frequency of the applied electric field and solution conductivity, for the two main situations where the field gradients are generated using either conductive electrode structures (eDEP) or posts/constrictions fabricated from insulator materials (iDEP) [14,15,16,17,18,19,20,21,22,23,24,25,26,27,28,29,30,31,32,33,34,35,36,37,38,39,40,41,42]. Most investigators observed positive DEP, but in two cases negative iDEP is reported [17,25]. Cao et al. [26] observed a transition from positive to negative DEP, with the cross-over frequency located between 1 and 10 MHz. The DEP results obtained [2,16,18,24,25,26,27,28,29,30,31,32,33,34,35,36,37,38,39,40,41,42] for proteins other than BSA are summarized in Figure 5. Most research groups report positive DEP for frequencies up to 6 MHz (including direct current), and of particular note is the observation that avidin and prostate specific antigen (PSA) exhibit a DEP cross-over frequency at ~10 MHz [27,37]. As summarized in Figure 6, various concentrations of BSA in aqueous solution have been employed. Included in Figure 6 are the concentrations used for streptavidin which, after BSA, is the most studied protein for its DEP characteristics. The mean separation distances between protein molecules for the various protein concentrations are also shown in Figure 6. This information is of relevance regarding any discussion of possible interaction between molecules. The molecular separation was estimated on the basis that a 1M solution contains Avogadro’s number of molecules—i.e., 0.6 molecules/nm^3^. The volume occupied per molecule is thus 1.66/C nm^3^ for a C molar solution. The separation distances shown in Figure 6 were calculated by taking the cube root of the volume per molecule. A wide range of values for the field gradient factor ∇*E*^2^ has been reported by the various investigators for a range of proteins, or has been estimated by Hayes [5] in his review. These values are shown in Figure 7 for both iDEP and eDEP studies, together with an indication of the minimum value of $\nabla E_{m}^{2}$ (3.5 × 10^21^ V^2^/m^3^) calculated according to Equation (9), required to attain a DEP force that overcomes the dispersive forces of Brownian motion. The adjusted minimum value (~4 × 10^18^ V^2^/m^3^) based on the empirical relationship described in Section 3.4 is also shown in Figure 7.

### 3.2. Bovine Serum Albumin (BSA) 

BSA is a well-studied, water soluble, protein. It has a molecular weight of 66.5 kDa; is composed of 583 amino acid residues (54% of which form six α-helices); takes the form of a prolate ellipsoid of dimensions 14 nm × 4 nm × 4 nm; has an isoelectric point in water of 4.7 at 25 °C [43,44]. Most DEP experiments on BSA have employed pH buffers to maintain a pH of 7.4 or higher. For such studies the protein molecules were thus negatively charged (the number of ionized acidic side-groups exceeded that of basic ones). Unbuffered solutions of 0.1 mM concentration and lower typically have a pH of 5.0–6.0 and are less negatively charged, close to having an equal number of, but not uniformly distributed, ionized acidic and basic groups. The BSA monomer contains 17 disulphide bridges between adjacent cysteine groups of its polypeptide chain, whilst bonding of the one free cysteine (Cys34) between interacting monomers leads to the formation of a dimer. BSA adopts its normal globular form between pH 4.5 and 7.0, but partially unfolds as the pH approaches the range 8.0–9.0 [45,46]. This unfolding involves the breaking and rearrangement of disulphide bonds, which is temperature sensitive and can lead to a loss of α-helix content [47,48] and irreversible self-aggregation [49]. It should also be noted that monomer, dimer and other aggregates typically exist in commercial samples of BSA [50]. Proteins in general follow first and second order aggregation kinetics [51]. Conformational change is the rate limiting step in the first order kinetics, making the rate of reaction independent of initial protein concentration. The second order reaction rate does depend on concentration, because molecular collision frequency limits formation of dimers, trimers, etc., and heat-induced aggregation. The suggestion by Nakano et al. [19] that ‘most iDEP manipulations of proteins may require the control of protein aggregation’ is well-founded, and as discussed in Section 3.4 may be of relevance to understanding the two conspicuous cases [17,26] of negative iDEP observed for BSA (Figure 3). 

### 3.3. The Dielectric β-Dispersion

Of particular relevance to protein DEP is the fact that globular proteins possess an intrinsic dipole moment. The magnitude of this moment is given by the resultant of the moments of the amino acids in the polypeptide chain (especially the additive effect of those forming α-helices), the moments of the charged acidic and basic groups about the molecule’s hydrodynamic center, and polarizations of the surrounding water molecules [52]. If the protein molecule is free to rotate about its prolate major and minor axes, this dipole moment manifests itself as a large dielectric dispersion (known as the β-dispersion)—the form of which for BSA is shown in Figure 8. By analyzing this dispersion, Moser et al. [53] computed dipole moment values for the BSA monomer and dimer as 384 D (1.28 × 10^−27^ Cm) and 636 D (2.12 × 10^−27^ Cm), respectively. The angle between the dipole moment and the long axis of the monomer was determined to be 50°. Moser et al. [53] performed dielectric and transient birefringence measurements on BSA solutions of concentrations in the range 0.2–1.4 mM and observed the effect of strong intermolecular interactions. In their measurement of the β-dispersion, Grant et al. [54] considered that the BSA concentrations (0.6–5.5 mM) were “high enough to permit molecular interaction”. 

### 3.4. Empirical Relationship Connecting Clausius–Mossotti (CM) and the β-Dispersion

For dielectric and impedance spectroscopy measurements on cell suspensions of sufficiently low volume concentrations *c*_v_, the dielectric increment Δ*ɛ* depicted in Figure 8, as well as the conductivity increment Δ*σ* characterizing this dispersion in terms of the increase of conductivity of the suspension with increasing frequency, are given by: (10)$$\Delta \varepsilon =3c_{v}\varepsilon_{m}CM;\;\Delta \sigma =\frac{1}{\tau }\Delta \varepsilon$$

The relationship between Δ*ɛ* and Δ*σ* results from application of the Kramers–Kronig transforms, where *τ* is the characteristic relaxation time of the Maxwell-Wagner interfacial polarization giving rise to the β-dispersion [4] (Chapter 9). Equation (10) can be extended to accommodate larger values of *c_v_* and to derive multi-shell models for analyzing impedance and DEP measurements on cell suspensions [4]. However, as stated elsewhere [9] (without the following explanation), Equation (10) is not applicable to protein suspensions. According to the Maxwell–Wagner mixture theory for particle suspensions, the measured effective permittivity *ε_eff_* of a dilute particle suspension is given by [4] (pp. 222–223): (11)$$\frac{\varepsilon_{eff}-\varepsilon_{m}}{\varepsilon_{eff}+2\varepsilon_{m}}=c_{v}\frac{\varepsilon_{p}-\varepsilon_{m}}{\varepsilon_{p}+2\varepsilon_{m}}$$
with *ε_p_* and *ε_m_* the particle and medium relative permittivity, respectively. The term effective permittivity is used to signify that a defined volume of a particle suspension may be replaced conceptually with an equal volume of a homogeneous medium of smeared-out bulk properties. 

Substitution of one volume with the other is assumed to not alter the electric field in the surrounding medium. The assumption is thus made that *ε_eff_ ≈ ε_m_*, implying that for a sufficiently large observation scale a heterogeneous compound material can be considered as a homogeneous one. Inserting this approximation into the denominator of the left-hand side of Equation (11) leads to the expression for Δ*ɛ* in Equations (10). However, an instructive result is obtained if this is applied to form a relationship between the Clausius–Mossotti factor *CM* and the dielectric increments depicted in Figure 8. For a dilute protein suspension, this relationship should thus be of the form: (12)$$CM=\frac{\Delta \varepsilon }{3c_{v}\varepsilon_{m}}=\frac{\Delta \varepsilon }{3\varepsilon_{m}}\left(\frac{C_{w}\rho_{p}}{C_{p}\rho_{w}}\right)$$
where *C_w_* and *C_p_*, *ρ_w_* and ρ_p_, respectively, are the molar concentration and mass density of pure water and the protein, respectively. The concentration *C_w_* of pure water is taken as 55.5 M (1000 g/L divided by its molecular weight of 18 g/mol), and protein density values can be derived using the molecular-weight-depending function derived by Fischer et al. [55]. Equation (12) qualitatively predicts that in a frequency range where there is a dielectric increment $\Delta \varepsilon^{+}$ a positive value for *CM* and a positive DEP response will result. As indicated in Figure 8 the opposite case should also hold for the high-frequency range where a dielectric decrement is exhibited. But can the *CM*-factor of Equation (12) simply be inserted into Equation (5) to describe protein DEP? Moser et al. [53] obtained $\Delta \varepsilon /C_{p}$ = 1.11 per mM for a BSA monomer concentration, so that with *ε_m,_* = 78.4 and *ρ_p_* = 1.41 gm/cm^3^, Equation (12) yields the result *CM* = 369. This is not possible according to the definition and limited range of values (1.0 > *CM* > −0.5) of the macroscopic *CM* factor derived from Equation (2) for the induced polarization *P_p_* per unit volume of a particle. Furthermore, based on the work of Takashima and Asami [56] who obtained values for Δε (per mM protein concentration) of 5.06 and 37.24 for cytochrome-c and carboxypeptidase, respectively, the corresponding values obtained for *CM* are 1745 and 12,480, respectively! This is the basis for stating [9] that the macroscopic theory leading to Equation (10) cannot be employed at the molecular level. 

If, instead of the assumption *ε_eff_* ≈ *ε_m_*, the identity *ε_eff_ = κ**ε_m_* is inserted into the denominator of the left-hand side of Equation (11) we obtain the relationship: (13)$$\left(\kappa +2\right)CM=\frac{\Delta \varepsilon }{\varepsilon_{m}}\left(\frac{C_{w}\rho_{p}}{C_{p}\rho_{w}}\right)$$

Values for the parameter (*κ* + 2)*CM* are given in Table 1, based on values of $\Delta \varepsilon /C_{p}$ obtained experimentally [56,57,58,59,60,61,62] for a range of globular proteins. No obvious relationship can be seen to link the value of a protein’s effective polarization factor (κ + 2)*CM* (per unit volume) with its molecular weight. Based on Equations (10) and (13) and the data given in Table 1, the following empirical relationship is proposed that links the molecular- (micro-) and macro-scales:*CM*_micro_ = (*κ* + 2)*CM*_macro_(14)

For the DEP of macro-particles, such as mammalian cells and bacteria, the plane of hydrodynamic shear of the particle, as it undergoes DEP through its suspending medium, can be considered to coincide with its ‘mathematical’ boundary at the particle–medium interface. At the molecular scale applicable to protein DEP, however, the situation is far more complicated. The plane of shear is most likely to lie within the outer boundary of the protein’s hydration shell, whose total extent is defined when the protein is stationary. We have, as shown schematically in Figure 9, the equivalent of a molecular ‘Russian doll’. The protein with its permanent dipole moment and most strongly ‘attached’ water that can rotate with it, occupies the inner cavity. The protein’s dipole field extends beyond an outer ‘macroscopic’ boundary at which the macroscopic boundary conditions of classical electrostatics can be applied. The medium polarization *P**_m_* must be uniform right up to this boundary. Located within this mathematical boundary is the hydrodynamic plane of shear (defining the zeta-potential determined by electrophoresis) and the protein’s outer hydration sheath. It is tempting to propose a conceptual equivalence of Equation (14) in terms of the ratio of two polarizations and interfacial dipole moment free energies: (15)$$\frac{P_{i}}{P_{m}}\equiv \frac{\chi_{i}E_{i}}{\chi_{m}E_{m}}\equiv \frac{\langle M_{i}\rangle \cdot E_{i}}{\langle M_{m}\rangle \cdot E_{m}}\propto \left(\kappa +2\right)CM_{\mathrm{macro}}$$
where suffices *i*, *m* identify the polarization, susceptibility, induced moment and local field in the protein cavity and bulk medium, respectively. These ratios will be sensitive to the physico-chemical attributes of a particular protein (e.g., peptide chain folding, net charge and the distribution of polar and hydrophobic groups on the protein surface) and could explain the very wide range of values of the parameter (*κ* + 2)*CM* given in Table 1. At this stage it is of interest to note that the large values given for ribonuclease (7000–11,000) and concanavalin (~15,000) would place these proteins above the minimum required level indicated in Figure 7 for BSA. DEP measurements for the other proteins cited in Table 1 would be of considerable value in this speculative argument. 

### 3.5. The β-Dispersion and Dipole Moment Density 

The β-dispersion can also conceptually be linked to the DEP frequency response of the BSA monomer in terms of the polarization (dipole moment density) of the medium and protein molecule. Two approaches can be adopted. The first involves the ‘book-keeping’ exercise of calculating the change Δ*U* of free energy stored in the field as a result of the following three actions: (i) Increase the field *E_m_* from zero (where *D_m_* = 0) to its final value (*D_m_* = *ɛ_o_ɛ_r_E_m_*) in the medium, that has a total volume *V_m_*; (ii) reduce *D_m_* by removing from the medium a cavity of volume *v_p_* large enough to contain the hydrated protein molecule; (iii) account for the incremental change (either positive or negative) of the medium polarization resulting from its interaction with the field of the protein’s induced and permanent dipole moment. These three actions can be expressed in the form [4] (pp. 87–89): (16)$$\Delta U=\frac{1}{2}\int_{V_{m}}\int_{0}^{D}E_{m}\cdot \delta Ddv-\frac{1}{2}\int_{v_{p}}E_{m}\cdot D_{m}dv-\delta U$$

Volume *V_m_* is very much larger than *v_p_* and so the first integral in Equation (16) represents a significantly larger contribution to *ΔU* than the second integral. The *δU* term thus plays a significant role. In the macroscopic derivation of the Maxwell–Wagner mixture theory that leads to Equation (10) the assumption is made that *ε_eff_* ≈ *ε_m_*. This effectively removes the requirement for calculating the *δU* term in Equation (16), which at a molecular scale is a significant weakness. Evaluation of *δU* can conceptually, for our present purpose, be accomplished by assuming the applicability of the boundary condition regarding continuity of the normal component of displacement flux (*D = ɛ_r_ɛ_o_E_m_*) across the interface between the solvated protein and the bulk medium. The free energy change *δU* is then given by an integral of the following form [4] (p. 89), taken over the protein’s effective cavity volume *υ_p_*: (17)$$\delta U=\frac{1}{2}\int_{\nu_{p}}\left(\varepsilon_{m}-\varepsilon_{p}\right)E_{i}\cdot E_{m}dv$$

For the frequency range where the dielectric increment *Δɛ*^+^ has a finite value in Figure 8, the protein’s effective permittivity *ε_p_* can be regarded as being greater than *ε_m_*. The integral in Equation (14) thus yields a negative value for *δU*. According to the work-energy theorem, for frequencies lower than *f*_xo_, work will be required on the particle by the field to withdraw it from the medium. Furthermore, this free energy is further reduced if the field *E_m_* increases. The protein monomer or dimer will attempt to minimize its electrostatic free energy by moving up a field gradient to a maximum value of this gradient. This describes the action of positive DEP. For frequencies lower than *f*_xo_ (i.e., where the dielectric decrement *Δɛ*^−^ has a finite value), the protein’s effective permittivity is less than that of the medium. The protein will move down a field gradient to search for a field minimum. Work is required by the field to insert the protein into the medium. This describes negative DEP. It is tempting to consider the cross-over of DEP polarity at 1–10 MHz for BSA, observed by Cao et al. [26], as experimental evidence for this scenario, because such cross-over is expected from inspection of the β-dispersion shown in Figure 8.

A second approach to linking the β-dispersion to protein DEP is to consider the time-averaged potential energy of the polarized protein particle in terms of its polarizability α per unit volume in unit field [4] (p. 89): $\langle U\rangle =-\frac{1}{2}\alpha E_{m}^{2}$ (per unit volume). From the fundamental relationships between the fields *E*, *D*, *P* and the dipole moment *M* per unit volume (see Figure 1) we have the following expression for *δU* in terms of the surface polarization *P* and induced dipole moment *M_p_* of the solvated protein: (18)$$\delta U=\frac{1}{2}\langle M_{p}\rangle \cdot E_{m},\;\mathrm{where}\;M_{p}=\int_{v_{p}}P_{s}\cdot \hat{n}dv$$

The magnitude of *M_p_* will give the strength of the DEP force, whilst its polarity will also define the *F_DEP_* polarity. A negative value for *M_p_* will indicate it is directed against the direction of *E_m_*. Work will be required to insert the polarized particle into the field *E_m_* within the aqueous medium. This describes negative DEP. A polarized protein possessing a positive value for *M_p_* will be aligned with *E_m_* and exhibit positive DEP. 

Defining the protein’s effective cavity volume *v_p_* to be used in the integrals of Equations (17) and (15) is not straightforward. Different protein molecules have from 0.20 to 0.70 g strongly associated (bound) water per g protein, contributing to its effective radius of rotation by up to one to two water molecule diameters [63]. From their studies, Moser et al. [53] determined a hydration of 0.64 g of H_2_O per g of BSA. Grant et al. [54] confirmed the existence of a subsidiary dispersion (δ-dispersion) in the frequency range 200–2000 MHz, and concluded that this dispersion is probably due to the rotational relaxation of water ‘bound’ to the protein. The term ‘bound water’ is taken to mean water bound to the protein by bonding of greater strength than the water–water bonding that exists in pure bulk water. This characteristic water structure that is formed near the surfaces of solvated proteins arises not only through hydrogen bonding of the water molecules to available proton donor and proton acceptor sites on the protein surface, but also through electrostatic forces associated with the water molecule’s electric dipole moment. The protein molecule and the water around it thus form a strongly coupled system, involving mechanical damping of the protein motion by adsorbed water, together with a dynamic electrical coupling between the tumbling electric dipole of the protein and the fluctuating dipoles of the adsorbed and bulk water. With such heterogeneity of the dielectric medium and also possibly of *E_i_* within the effective volume *υ_p_*, computation of the integrals in Equations (17) and (18) thus involves some ‘interesting’ challenges. Not least of which is defining the effective volume *υ_p_* of the protein, and how the normal components of displacement flux *D* and polarization *P* vary within the heterogeneous boundary between the protein’s surface and the bulk aqueous medium. 

### 3.6. Interfacial Polarizations

The formation of defect dipoles in both amorphous and crystalline polymers is known to influence their dielectric properties [64]. Examples of possible relevance to protein DEP are depicted in Figure 10. These are suggested examples where the standard boundary conditions of Maxwell-based electrostatics may not apply—the implications of which have been described by Martin et al. for the specific case of a ‘Rossky cavity’ [65]. The example shown in Figure 10a could, for example, depict the disruption of the network of hydrogen bonds at a protein–water interface—possibly resulting in the creation of nanodomains that have the capability of dynamically freezing into a ferroelectric glass [66]. Ferroelectric materials are known to develop structures with curls on their faces where the field is no longer conservative [67]. This of relevance to Equation (4) in which *E**_m_* is assumed to be irrotational. Boundaries of the form depicted in Figure 10b between dielectrics of different permittivity have been shown, through theory and classical molecular dynamics simulations of hydrated cytochrome c, to exist in the hydration shells of proteins [68]. The large dispersion strength (*Δɛ* ~ 2400) shown in Figure 10c for a suspension of polystyrene microspheres was analyzed and determined not to arise from classical Maxwell–Wagner interfacial polarization, electrophoretic particle acceleration, or the presence of a frequency-independent surface conductance [69]. The most likely origin was considered to be a frequency-dependent surface conductance that varies with the ionic strength of the suspending aqueous electrolyte. Interfacial polarizations of these types should be included in the exercise to find a molecular-based DEP theory. It is also pertinent to mention that excised samples of biological tissue can exhibit large *Δɛ* values [52], a good example being skeletal muscle with measured relative permittivity *ε_r_* ≈ 10^7^ at 10 Hz [70]. This is known as the α-dispersion and, according to the convention used in assigning Greek letters to dielectric dispersions, occurs in a frequency range below that of the β-dispersion. 

### 3.7. Protein Dipole Polarization

Other paths to formulation of the DEP force acting on a protein permanent dipole are through either Equation (4) or, as follows, the relationship between *U_DEP_* and *F_DEP_* given by Equation (7). In the absence of an electric field, the orientations of the dipole moments of proteins in solution will on average be distributed with the same probability over all directions. On average their net dipole moment in any specific direction is zero. On application of a field each dipole will experience a field alignment torque *m* × *E_i_*, so that net polarization results. The electrical potential energy *U* of each dipole is given by *U = −(mE_i_*cos*θ*), where *θ* is the angle between the dipole moment and the local field vector *E_i_*. From Boltzmann–Maxwell statistics the probability of finding a dipole oriented in an element of solid angle *dΩ* is proportional to exp(−*U/kT*), with *k* the Boltzmann constant and *T* in kelvin. A moment pointing in the same direction as *dΩ* has a component (*m*cos*θ*) in the direction of *E_i_.* As detailed elsewhere [4,9] the thermal average of cos*θ* is given by the derivation of the so-called Langevin function: (19)$$\langle cos\theta \rangle =\frac{\int \;exp\left(-U/kT\right)cos\theta d\Omega }{\int \;exp\left(-U/kT\right)d\Omega }=\frac{m}{3kT}\left(1-\frac{1}{15}{\left(\frac{mE_{i}}{kT}\right)}^{2}\right)$$

For a monomer BSA dipole moment of *m* = 384 D and *E_i_* ≈ 3 × 10^5^ V/m (e.g., Lapizco-Encinas et al. [17], assuming *E_i_* ≈ *E_o_*) the factor (*mE_i_/kT*) ≈ 0.01. So, to a good approximation $m\langle \cos\theta \rangle =m^{2}E_{i}/3kT$. For *E_i_* > 3 × 10^7^ V/m (e.g., Cao et al. [26]) the full expression for the thermal average of cos*θ* should be used. With an applied field less than 10^6^ V/m, then through Equation (7) the average orientational DEP force (*F_oDEP_*) acting on a protein’s dipole is given by: (20)$$F_{oDEP}=-\nabla U_{DEP}=\left(m\langle \cos\theta \rangle \cdot \nabla \right)E_{i}=\frac{m^{2}}{3kT}\left(E_{i}\cdot \nabla \right)E_{i}=\frac{m^{2}}{6kT}\nabla E_{i}^{2}$$

This expression for *F_oDEP_*, which also follows from Equation (4), has two important features. The first is that the DEP force exerted on a polarized protein molecule possessing a permanent dipole moment is directly proportional to ∇*E*^2^. Previously, one of the authors [9] has concluded that for frequencies below *f*xo (see Figure 8) proteins with a permanent dipole should exhibit positive DEP directly proportional to ∇*E*, and not ∇*E*^2^, whereas negative DEP should be expected solely above *f*xo and have a ∇*E*^2^ dependence. This conclusion is only valid for a ‘rigid’ protein molecule whose dipole is constrained from responding to the alignment torque *m*×*E_i_,* or where the relaxation time of the protein’s permanent dipole is too slow to respond to a high-frequency oscillating field. Based on Equation (20) the ratio of the DEP force exerted on an orientationally polarized dipole moment to that on an induced dipole moment (Equation (5)) is: (21)$$\begin{array}{c} \frac{F_{oDEP}\left(orientation\right)}{F_{DEP}\left(induced\right)}=\frac{m^{2}}{12\pi R^{3}kT\varepsilon_{o}\varepsilon_{m}\left[CM\right]}=1.85\times 10^{28}\frac{m^{2}}{R^{3}}\frac{E_{i}}{E_{m}}\\ \left(k=1.38\times 10^{-23}\;J\cdot K^{-1};\;T=300\;K;\;\varepsilon_{m}=80;\;CM=0.5\right) \end{array}$$

For monomer BSA (*m* = 1.28 × 10^−27^ Cm; *R* = 3.5 nm), and assuming *E_i_* = *E_m_*, this gives near equality of *F_oDEP_* and *F_DEP_* (*F_oDEP_* = 0.71 *F_DEP_*). This result indicates that unless *E_i_* >> *E_m_*, Equation (20) does not offer a theoretical basis to explain why the majority of experimental ∇*E*^2^ values shown in Figure 7 fall well below the minimum requirement of ∇*E*^2^ > 3.5 × 10^21^ V^2^/m^3^. It also implies that we require a better understanding of the relationship between the DEP force and the β-dispersion shown in Figure 8. Qualitatively, a protein molecule will exhibit positive DEP if the polarization per unit volume (i.e., total dipole moment) of the bulk water it displaces is less than that of the protein and its associated water molecules of solvation. A quantitative understanding should include a molecular-level description of short- and long-range interactions of the dipoles (protein–water and water–water) and the nature of the interfacial and/or dipole charges that can create the situation *E_i_* >> *E_m_*. A route to this might be offered through the suggested empirical relationship given in Equation (15), that relates the protein’s local cavity field and its polarization to the large values of the effective polarization factor (*κ* + 2)*CM* given in Table 1. Of the proteins listed in Table 1, only three (BSA, concanavalin, ribonuclease) appear to have been investigated for their DEP characteristics. It is of interest to compare the locations of these proteins in the ∇*E*^2^ ‘ranking’ of Figure 7, with their relative values of (*κ* + 2) *CM* given in Table 1 (~1000: BSA; ~11,000: ribonuclease; ~ 15,000: concanavalin). If the macroscopic *CM* factor is replaced by the proposed microscopic version (*κ* + 1)*CM* in Equation (9), then the DEP results cited for ribonuclease and concanavalin lie well above the ‘minimum required’ level in Figure 7. 

### 3.8. Protein Stability

Concerning the interesting cases [17,25] of negative iDEP indicated for BSA in Figure 3, both studies were carefully performed and analyzed, so there is no intent here to label their experiments as ‘wrong’. It is often the case in biological work that the ‘odd’ finding is the very one to pursue further. Lapizco-Encinas et al. [17]—the first to report protein iDEP—employed a BSA concentration of 0.46 mM, buffered at high pH (8 and 9) and ionic conductivities (10 mS/m). This brings their situation to within the bounds of protein conformational change and unfolding, as well as loss of α-helix content and self-aggregation [45,46,47,48,49]. A concentration of 0.46 mM is also within the range (0.2–0.6 mM) where dielectric studies [53,54] provided evidence of strong intermolecular interactions (see also Figure 6). As a general rule, in an aqueous environment with a high ionic strength (i.e., high conductivity) the solvated ions compete with the protein molecules in binding with water, to such an extent that the protein molecules tend to associate with each other. This is because protein–protein interactions become energetically more favorable than protein–solvent interaction [71]. The result is the precipitation of the least soluble solute—namely the protein. This could easily have been interpreted by Lapizco-Encinas et al. as collection of the protein by negative DEP. There is also the possibility that true iDEP of aggregates, rather than precipitation, was observed. This would explain why a very small field (~10^5^ V/m) could be employed, and might also provide insights into the DEP behavior of a test sample as it makes the transition from the molecular- to the macro-scale. In their studies, Zhang et al. [25] employed low sample concentrations (0.78 μM) but high conductivities (0.1 S/m). The likelihood of molecular interactions and self-aggregation was thus low (Figure 6) but with such a high ionic strength the precipitation of the BSA was likely. 

### 3.9. Other Experimental Details

Comprehensive details of electrode and chamber designs for both iDEP and eDEP devices have been reviewed elsewhere [5,7,72,73,74] and are not considered here. Also, for some studies thorough consideration may not have been given to the possible confounding influence of electrothermal effects. We consider these to be relatively minor considerations for the bigger picture. The following experimental aspects are, however, suggested for further consideration.

For a quantitative interpretation of the published results one has to be aware that the reported experimental parameters are often not given or might be somewhat uncertain. One reason for this is the high surface-to-volume ratio of the microfluidic system. This is required because microscope-aided observation of protein DEP calls for flat observation chambers with typical heights between 20 and 200 µm, ranging down to 2 µm [24,39] and even 200 nm [33,34]. This relatively large surface area can result in uncertainties concerning conductivity, pH value and solute concentration. At initially low ionic strength tiny amounts of contamination can lead to a substantial increase in conductivity. This holds, to a lesser extent, also for the pH value. Depending on the experimental arrangement, diffusion of CO_2_ from the environment can lead to an increased conductivity and lowered pH value. In DC-DEP, artificial pH gradients might also be generated in a way similar to the preparation of pH gradients for isoelectric focusing. Due to adsorption at the surface of the measuring chamber, as well as within fluidic tubing, solute concentrations can decrease even in the course of the actual experiment. Often, counter-measures are taken using buffers or surface modifications [19,32]. Published results should thus be compared and interpreted carefully. 

Another cause of uncertainty is the determination of electrical parameters. Sometimes it is not clear whether voltages are given as peak-to-peak or as root-mean-square (rms) values. In about half the work on protein DEP, values of either |*E*| or ∇ |*E*|^2^ are calculated. Both values are given for only a few of the studies cited here [16,20,38]. The spatial distribution of just |*E*| is given by Agastin et al. [18], that of ∇ |*E*|^2^ in rather more cases [20,22,26,28,32,37] and sometimes the distribution of both values is given [22,26,38]. Owing to experimental limitations actual measurements of |*E*| or ∇ |*E*|^2^ have not been carried out in any of these works. All calculations have been performed numerically by commercial software based on finite-element-methods (FEM). Although this is not specified by any of the authors, it is very probable that the spatial models of these simulations were based on simple geometrical bodies like cuboids and cylinders. This means that in essence the edges are modelled with infinitesimal radius of curvature. This should lead to infinite values of both |*E*| and ∇ ∇ |*E*|^2^ since both are calculated as spatial derivatives of the potential distribution. In practice, this is not the case because the calculations are performed on a mesh or grid with finite resolution. This means that the field distributions are qualitatively correct. However, the maximal values of |*E*| and ∇ |*E*|^2^ are now dependent on the spatial resolution of the mesh. It might well be that in several cases the resolution is not known because the software automatically adapts the mesh locally. In only two reports have the resolutions been given—namely, values of 50 nm³ [38] and 100 nm³ [23]. In order to determine the impact of the chosen resolution we have calculated the field distribution for two basic electrode arrangements, i.e., for co-planar interdigitated electrodes and for arrays of cylindrical pins. Using the FEM software Maze (Field Precision, Albuquerque, USA) the resolution of the Cartesian grid was varied from 120 nm down to 12 nm. This produced a roughly linear increase of both |*E*| and ∇ |*E*|^2^ (data not shown) with resolution (i.e., with the inverse of the linear voxel dimensions). For interdigitated electrodes |*E*| and ∇ |*E*|^2^ increase by a factor of 4 and 10, respectively, whilst for cylindrical arrays these factors amount to 2 and 60, respectively. As a consequence, the currently available data on |*E*| and ∇ |*E*|^2^ should only serve as a more or less rough estimate when comparing them with physical theory. 

## 4. Concluding Comments

Commencing with the first reported studies in 1994 of the DEP responses of avidin, chymotripsinogen, concanavalin and ribonuclease [2] at least 22 different globular proteins have now been investigated for their DEP responses [2,14,15,16,17,18,19,20,21,22,23,24,25,26,27,28,29,30,31,32,33,34,35,36,37,38,39,40,41,42]. Aspects of this work are examined here, covering details not encompassed in previous reviews [1,3,4,5,6,7] of protein DEP. Apart from a few cases, whether through insulator-based (iDEP), electrode-based (eDEP) investigations, at DC or with applied field frequencies ranging from 20 Hz to 30 MHz, the reported results are largely consistent. In their DEP analyses all the authors employ the standard induced-dipole moment theory that employs the Clausius-Mossotti (*CM*) factor derived from macroscopic electrostatics. However, apart from the three studies of Laux et al. [23], Zhang et al. [25] and Cao et al. [26], none of the reported DEP responses can be explained in terms of the limitations set by this classical theory. As shown in Figure 7, only these three studies employed a gradient field factor $\nabla E_{m}^{2}$ > 3.5 × 10^21^ V^2^/m^3^ required, according to Equation (9), to overcome the dispersive forces associated with the Brownian motion of the protein molecules. All of the other studies fell far short of this requirement. In one reported DEP manipulation of BSA, a value of 10^12^ V^2^/m^3^ is cited [17]. 

Of particular relevance to protein DEP is the fact that globular proteins possess an intrinsic dipole moment. If the protein molecule is not rigid, but free to rotate about a major or minor axis when subjected to an applied AC field, this dipole moment manifests itself as a large dielectric dispersion known as the β-dispersion. The form of this dispersion for BSA is shown in Figure 8. For the frequency range where the β-dispersion exhibits a dielectric increment *Δɛ*^+^, the protein’s effective permittivity *ε_p_* can be regarded as being greater than the value *ε_m_* for the surrounding medium. This should result in a positive DEP response. A negative DEP response should then be exhibited on increasing the field frequency to the part of the β-dispersion where a dielectric decrement occurs, as shown in Figure 8. There are three examples where a DEP cross-over (transition from positive to negative DEP with increasing frequency) has been observed at 1–10 MHz, namely: that reported for BSA by Cao et al. [26] as shown in Figure 3; for avidin (Bakewell et al. [27]) and PSA (Kim et al. [37]) as shown in Figure 5. This is consistent with the DEP responses of these proteins resulting from polarization of their permanent dipole moment, and not only as the result of an induced dipole moment. 

The DEP force arising from a permanent dipole moment is given by Equation (20), and is shown to be directly proportional to ∇*E*^2^. This corrects a previous conclusion [9], based on the presumption of a rigid rather than rotationally free permanent dipole, that the DEP force arising from a permanent dipole would be proportional to ∇*E*. However, as shown by Equation (21), the contribution of the DEP force expected for a BSA from its permanent dipole moment is predicted (according to current accepted theory) to be slightly less than the contribution of its induced moment. This indicates that, unless the ‘cavity’ field experienced by the protein molecule is very much larger than the field existing within the surrounding bulk medium, we have is no explanation in terms of the standard DEP theory (even if modified to encompass both an induced plus a permanent dipole moment) why the majority of experimental ∇*E*^2^ values shown in Figure 7 fall well below the minimum requirement of ∇*E*^2^ > 3.5 × 10^21^ V^2^/m^3^ to overcome thermal dispersion effects. As shown in Figure 7, the minimum required ∇E^2^ value is lowered by a factor of ~1000-fold for BSA, if the macroscopic *CM*-factor is replaced in Equations (5) and (9) by the empirically based molecular version *CM*_micro_ = (*κ* + 2)*CM*_macro_ formulated in Section 3.4, and tabulated for various proteins in Table 1. Of the proteins listed in Table 1, only three (BSA, concanavalin, ribonuclease) are cited in Figure 7. The location of these proteins in the ∇*E*^2^ ‘ranking’ of Figure 7 is significant. Their relative values of (*κ* + 2)*CM* given in Table 1, namely: ~11,000 for ribonuclease and ~15,000 for concanavalin, would place them above the minimum requirement level indicated in Figure 7 for BSA. It would clearly be of value to populate Table 1 with as yet unavailable dielectric spectroscopy data for the other proteins cited in Figure 5, and vice versa. With this information protocols could be developed to spatially manipulate or selectively sort targeted protein molecules, so bringing protein DEP in line with the achievements and promise enjoyed by the more established DEP of cells and bacteria, for example [75]. 

Finally, Equation (15) is offered for the following relationships between the ratios of the polarization of a protein in its cavity field and of the surrounding medium:
$$\frac{P_{i}}{P_{m}}\equiv \frac{\chi_{i}E_{i}}{\chi_{m}E_{m}}\equiv \frac{\langle M_{i}\rangle \cdot E_{i}}{\langle M_{m}\rangle \cdot E_{m}}\propto \left(\kappa +2\right)CM_{\mathrm{macro}}$$

These ratios will be sensitive to the physico-chemical attributes of a particular protein (e.g., peptide chain folding, net charge, and the distribution of polar and hydrophobic groups on the protein surface) and could explain the very wide range of values for the parameter (*κ* + 2)*CM* given in Table 1. This empirical-based suggestion mirrors various theoretical findings of Matyushov and co-workers [8,10]. The possible significance of this for further development of a robust theory for protein DEP is discussed in an accompanying paper [11]. 

## Figures and Tables

**Figure 1 micromachines-11-00533-f001:**
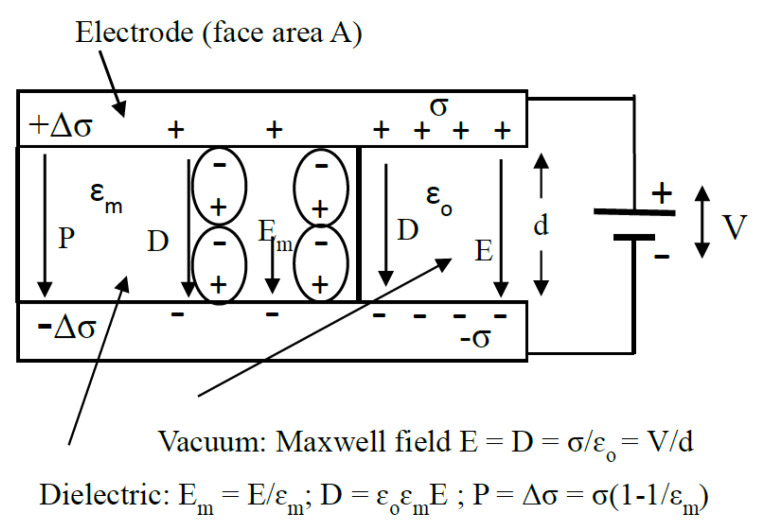
A dielectric of relative permittivity *ɛ_m_* is shown partly inserted between two electrified electrodes. ‘Free’ charge density *σ* on the electrodes creates the Maxwell field *E* and electric displacement *D* (both = *σ/ɛ_o_*). ‘Bound’ charge density Δ*σ* created by polarization (charge displacement) of the dielectric generates the polarization vector *P* ($\Delta \sigma =P_{s}\cdot \hat{n}=P$), and equates to the number density of polarized molecules—i.e., the dielectric’s dipole moment M per unit volume. These relationships give *D* = *E* + *P/ɛ_o_*, and *P* = *ɛ_o_(ɛ_m_ − 1)E_o_* = *χ_m_ɛ_o_E_o_*.

**Figure 2 micromachines-11-00533-f002:**
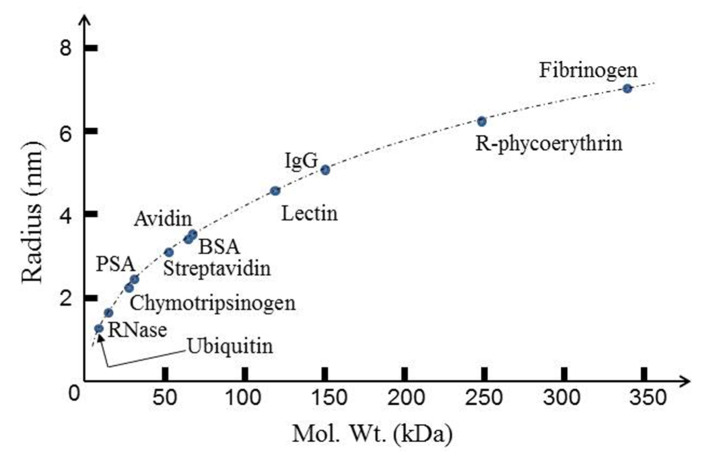
Globular proteins studied for their dielectrophoresis (DEP) response, with their hydrodynamic (Stokes) radii located on the empirical relationship between protein size and molecular weight (dotted curve) provided by Malvern Panalytical^®^ (Zetasizer Nano ZS).

**Figure 3 micromachines-11-00533-f003:**
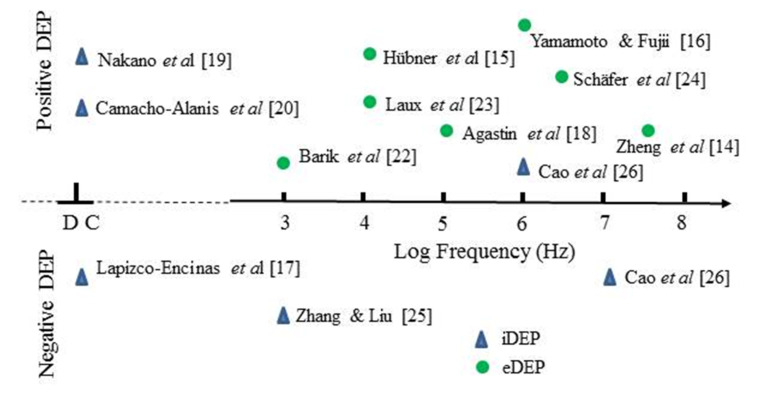
Insulator-based (iDEP) and electrode-based (eDEP) studies reported for bovine serum albumin (BSA). Most groups observed positive DEP, but two cases of negative iDEP have also been reported [17,25]. Cao et al. [26] report a DEP cross-over frequency between 1~10 MHz.

**Figure 4 micromachines-11-00533-f004:**
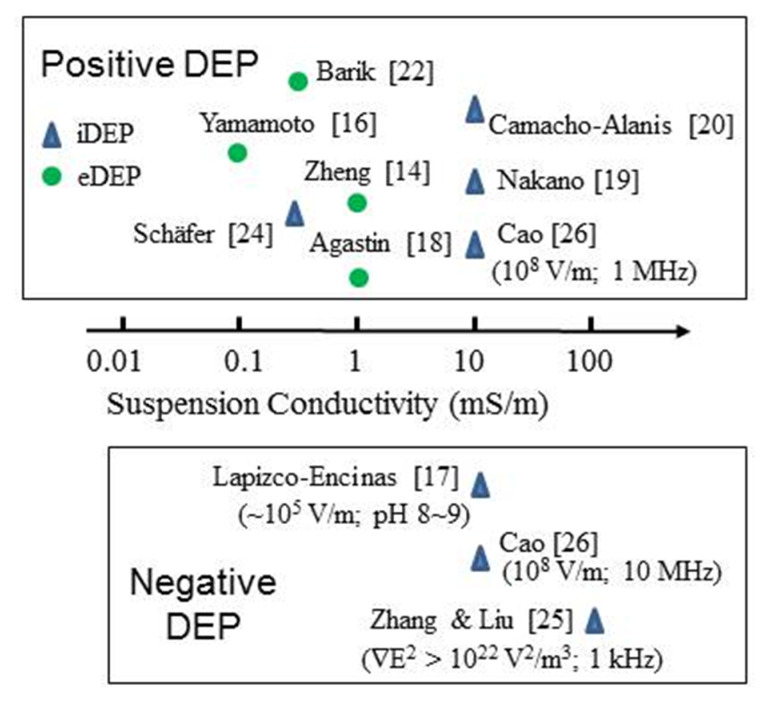
A range of aqueous solvent conductivity has been used in DEP studies of BSA. The experimental factors associated with the two cases [17,25] of negative iDEP and the cross-over of polarity between 1~10 MHz [26] are discussed in Section 3.2.

**Figure 5 micromachines-11-00533-f005:**
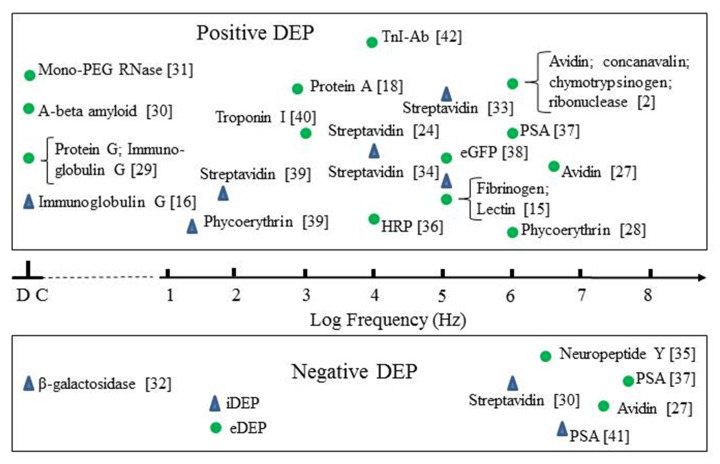
A summary of the DEP responses, at specific frequencies of the applied field, for proteins other than BSA. (HRP: horse radish peroxidase; PSA: prostate specific antigen; eGFP: enhanced green fluorescent protein; TnI-Ab: troponin I antibody).

**Figure 6 micromachines-11-00533-f006:**
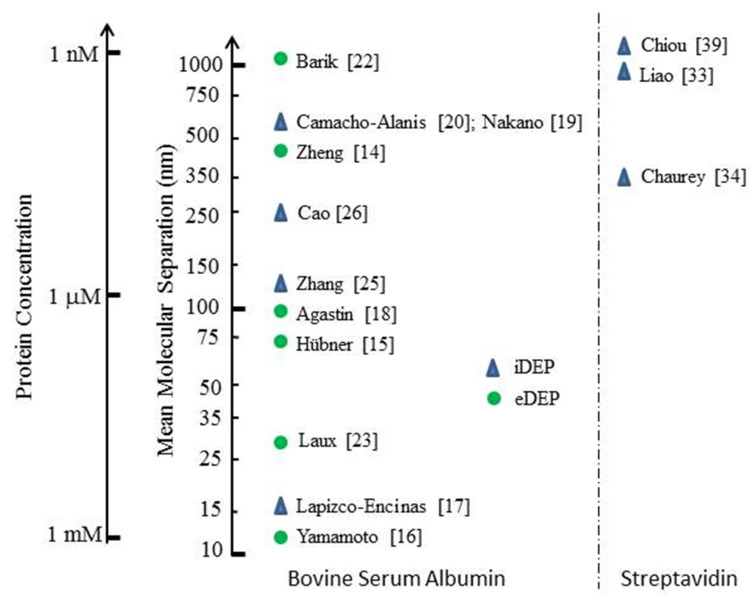
Mean distance between BSA and streptavidin molecules for the reported sample concentrations, estimated as the cube root of the volume occupied per protein molecule.

**Figure 7 micromachines-11-00533-f007:**
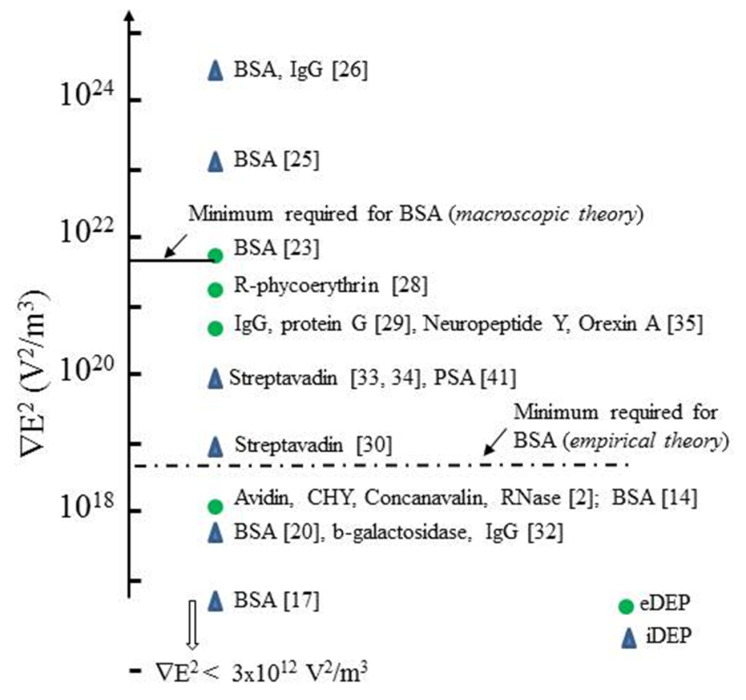
Values of the field gradient factor $\nabla E_{m}^{2}$ as reported by the investigators or estimated by Hayes [5]. The minimum value of $\nabla E_{m}^{2}$ required to compete against Brownian diffusive effects, calculated according to Equation (9), is shown for the case of BSA. The adjusted value shown for this is based on the empirical relationship described in Section 3.4.

**Figure 8 micromachines-11-00533-f008:**
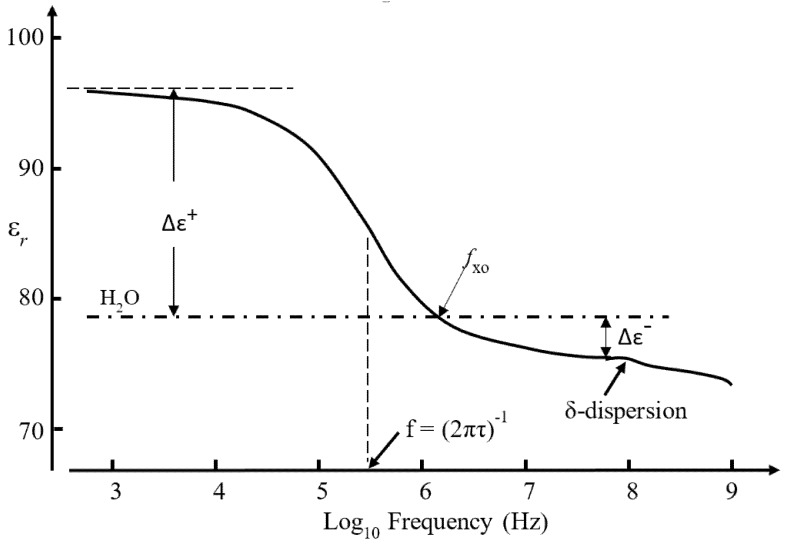
The β-dispersion and δ-dispersion, arising from orientation polarization of the protein and protein-bound water, respectively, exhibited by 0.18 mM BSA (based on Moser et al. [50] and Grant el al. [51]). The radian frequency of orientation relaxation for BSA is given by the reciprocal of its relaxation time τ. For frequencies below *f*xo (~1 MHz) the relative permittivity *ɛ_r_* of the BSA solution exceeds that of pure water, and is less than this above *f*xo. According to Equation (11) the dielectric increment $\Delta \varepsilon^{+}$ and decrement $\Delta \varepsilon^{-}$, respectively, specify the frequency ranges where positive and negative DEP, respectively, should be observed for monomer BSA in aqueous solution.

**Figure 9 micromachines-11-00533-f009:**
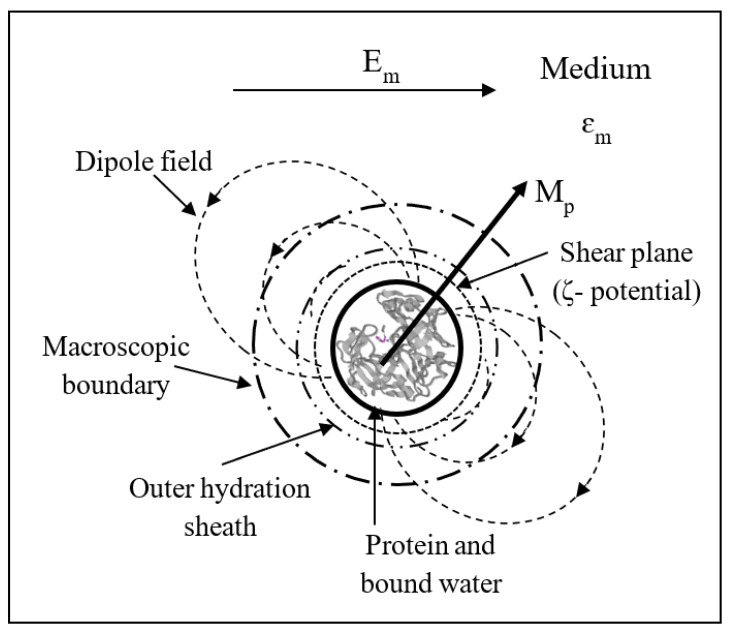
Schematic of a ‘Russian doll’ model for a protein with a permanent dipole moment M_p_, which occupies the innermost cavity together with its strongly bound water molecules. The protein’s dipole field extends beyond an outer macroscopic, ‘mathematical’, surface where the classical boundary conditions of electrostatics can be applied. Within this mathematical surface is a boundary that contains the protein’s outer hydration sheath, and the hydrodynamic plane of shear that defines the zeta-potential within the protein’s diffuse electrical double-layer.

**Figure 10 micromachines-11-00533-f010:**
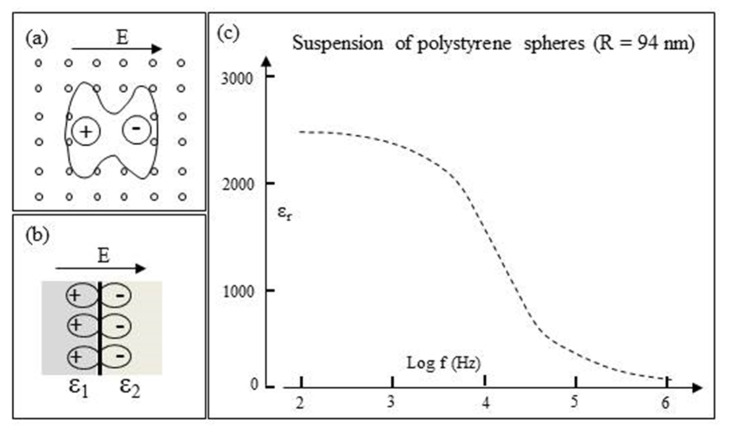
(**a**) Schematic of a dipole formed at the site of a structural defect in a molecular lattice. An example could be the disruption of the hydrogen bond network in bulk water at a protein–water interface—with the possible creation of ferroelectric nanodomains [66]. (**b**) Dipole polarization at a boundary of dielectric inhomogeneity. A solvated protein, with its bound water and surrounding bulk water, represents an inhomogeneous dielectric [68]. (**c**) Dielectric dispersion exhibited by an aqueous suspension of polystyrene nanospheres (R = 94 nm) (based on Schwan et al. [69]).

**Table 1 micromachines-11-00533-t001:** Values of the factor (κ + 2) *CM* given by Equation (13) for various globular proteins, derived from reported *Δɛ* and corresponding protein concentration values. The protein density values were derived from the weight-depending function given by Fischer et al. [55].

Protein	Mol. Wt.	Density (g/cm^3^)	*Δɛ/c_p_* (*c_p_*: mM)	(*κ* + 2)*CM* Equation (13)	Reference
Ubiquitin	8600	1.49	3.82	4020	[58]
RNAse SA	10,500	1.48	15.00	15,720	[57]
Phospholipase	13,000	1.46	1.82	189	[56]
Cytochrome-c	13,000	1.46	5.06	5240	[56]
Ribonuclease	13,700	1.46	11.0	11,400	[59]
			7.12	7350	[56]
Lysozyme	14,300	1.46	1.34	1390	[56]
Myoglobin	17,000	1.45	0.07	2090	[60]
			1.79	1440	[61]
Trypsin	23,000	1.43	6.74	6810	[56]
Carboxypeptidase	34,000	1.42	37.24	37,440	[56]
Hemoglobin	64,000	1.41	1.29	1290	[62]
BSA	66,000	1.41	1.11	1110	[53]
Concanavaline	102,000	1.41	15.31	15,270	[56]

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
