# Peer review of "Protein Dielectrophoresis: I. Status of Experiments and an Empirical Theory"

_micromachines, 2020, doi:10.3390/mi11050533_

Round 1
Reviewer 1 Report
This paper reviews the protein dielectrophoresis(DEP) and insights the difference between micro and macro CM factors. I think this content is very important for understanding the protein DEP. However, there are some unclear points and mistypes(?), so I recommend this paper needs minor revision.
The comments in content are as follows:
Whole
Use italic font for physical quantities in sentences and equations in figs.
L229, 230 E2
I think this is grad E^2
Figure 8, 9
The axis in graphs should be located outside the plot regions.
LL335-340
I think the content here is important, so I recommend inserting a schematic image for clear understanding.
LL343-344
This sentence is unclear, please improve. What is j?
LL515, 516, 630
Is this (k+2)CM?
LL562, 564
The expression of grad E^2 should be unify. ∇E2
L568
Specified -> specified
LL576-580
A space needs between the number and units. Ex) 50 nm^3
Author Response
Dear Reviewer
We are grateful for your very careful reading of our manuscript, and your valued suggestions for improvement.
We have made every change you have suggested - including a new figure (Fig 9) to describe the 'Russian Doll' model; changing the font in all equations and in the main text; altering the position of the axes in Figure 8 and in Figure 9 (now Fig 10).
A paragraph has been added to describe the new figure - and this is marked in our revised manuscript.
We have also corrected all of the finer details you have high-lighted (including ones we have discovered ourselves) - but these small changes have not been high-lighted in the main text.
Thank you.
Reviewer 2 Report
This paper presents an extensive review of research on protein dielectrophoresis and interesting empirical derivations to validate the current theoretical understanding. This work helps to further the existing knowledge and understanding in this field through the insights presented by the authors. No revisions are suggested for the paper.
Author Response
Thank you for your positive comments.
You have requested no changes to be made.
Reviewer 3 Report
Can be accepted in present form.
Author Response
Thank you for your positive remark.
You have requested no changes.